# Preoperative Differentiation of Uterine Leiomyomas and Leiomyosarcomas: Current Possibilities and Future Directions

**DOI:** 10.3390/cancers14081966

**Published:** 2022-04-13

**Authors:** Klaudia Żak, Bartłomiej Zaremba, Alicja Rajtak, Jan Kotarski, Frédéric Amant, Marcin Bobiński

**Affiliations:** 1I Chair and Department of Oncological Gynaecology and Gynaecology, Student Scientific Association, Medical University of Lublin, 20-059 Lublin, Poland; 54937@student.umlub.pl (K.Ż.); 55415@student.umlub.pl (B.Z.); 2I Chair and Department of Oncological Gynaecology and Gynaecology, Medical University of Lublin, 20-059 Lublin, Poland; alicjarajtak@umlub.pl (A.R.); jan.kotarski.gabinet@gmail.com (J.K.); 3Department of Oncology, Gynecological Oncology, KU Leuven, 3000 Leuven, Belgium; f.amant@nki.nl; 4Centre for Gynecologic Oncology Amsterdam (CGOA), Antoni Van Leeuwenhoek-Netherlands Cancer Institute (Avl-NKI) and University Medical Centra (UMC), 1066 CX Amsterdam, The Netherlands

**Keywords:** leiomyosarcoma (LMS), leiomyoma (LM), uterine fibroids, preoperative differentiation

## Abstract

**Simple Summary:**

Uterine sarcomas are the second most common unexpected malignancy diagnosed after surgery. It is worrisome, as its preoperative diagnosing can impact the choice of the treatment method, including surgery. Therefore, nowadays, many researchers are trying to find innovative methods to differentiate benign and malignant lesions of the uterus preoperatively. The review of the current literature showed that the use of more than one parameter in specific diagnostic scales is one of the most effective methods. Moreover, machine learning models and artificial intelligence (AI) are hope-giving directions, which may help in preoperative ULMS and ULM distinguishment. In order to collect a large amount of ULMS patients, multicenter databases seem necessary.

**Abstract:**

The distinguishing of uterine leiomyosarcomas (ULMS) and uterine leiomyomas (ULM) before the operation and histopathological evaluation of tissue is one of the current challenges for clinicians and researchers. Recently, a few new and innovative methods have been developed. However, researchers are trying to create different scales analyzing available parameters and to combine them with imaging methods with the aim of ULMs and ULM preoperative differentiation ULMs and ULM. Moreover, it has been observed that the technology, meaning machine learning models and artificial intelligence (AI), is entering the world of medicine, including gynecology. Therefore, we can predict the diagnosis not only through symptoms, laboratory tests or imaging methods, but also, we can base it on AI. What is the best option to differentiate ULM and ULMS preoperatively? In our review, we focus on the possible methods to diagnose uterine lesions effectively, including clinical signs and symptoms, laboratory tests, imaging methods, molecular aspects, available scales, and AI. In addition, considering costs and availability, we list the most promising methods to be implemented and investigated on a larger scale.

## 1. Introduction

Despite the efforts of many researchers, uterine sarcomas are still one of the most difficult lesions to diagnose preoperatively. This is especially true when compared to endometrial cancer, in which the use of endometrial biopsy allows us to diagnose 90% of patients [1]. As such, twenty years ago it was reported that up to 80% of patients who undergo surgery for leiomyoma have been referred inappropriately on the basis of ‘suspected malignancy’, meaning in our understanding ‘suspected sarcoma’ [2]. Lately, Yildiz et al. showed that among unexpected gynecological malignancies in hysterectomies carried out for benign indications the second most common malignancy was uterine sarcoma (7/38, 18%) [3]. As we know, ULMS is one of the rarest lesions of the uterus. In the study by Multinu et al. five unexpected sarcomas (including 4 ULMS) were diagnosed among 3759 hysterectomies for benign indications (0.13%), which is not seen as a great number. The incidence of sarcomas is slightly higher among surgeries for uterine fibroids and it amounts to 0.39% [4].

Nowadays, most of these procedures, especially those when LM is suspected, are performed using minimally invasive surgery [5]. This type of surgery is commonly associated with the use of morcellation, which allows the removal of solid masses without the need for an open surgery [6]. Nevertheless, in case of undiagnosed malignances this type of treatment may result in difficulties in pathological staging and the risk of local spread and recurrence, which is associated with worsened prognosis [7,8,9]. Therefore, the Food and Drug Administration (FDA) recommends using morcellation in the tissue containment system, although only for appropriately selected patients [10]. Before qualifying for a morcellation procedure, patient triage, based on tumor biology and sonographic characteristics should be performed [8]. Additionally, we should not forget that it is more difficult for pathologists to perform proper disease staging on morcellated tissue [11].

Hence, more and more researchers are trying to find the best preoperative methods, which can help in making ULMS and ULM diagnosis, and may become the new gold standard. Unfortunately, most known preoperative methods have not been validated and need further research. Thus, essential questions and doubts associated with this problem arise here; are the preoperative methods effective enough to get proper diagnosis preoperatively? This paper makes an attempt to answer this question.

## 2. Clinical Presentation

### 2.1. Specific Symptoms or Signs

No specific symptoms or signs that help to distinguish ULMS from ULM have been determined, so far. Some common symptoms, like uterine bleeding, palpable pelvic mass and pelvic pain are characteristic not only of ULMS, but they also appear in ULM. Köhler et al. proposed a preoperative ULMS risk score with the aim to facilitate diagnosis, drawing attention to bleeding symptoms: intermenstrual bleeding; hypermenorrhea; dysmenorrhea; postmenstrual bleeding; rapid growth. Nonetheless, the scale has not been effective without the usage of imaging characteristics: suspicious sonography and the tumor diameter. This scale is described in Section 5, “Available Scoring Scales” [12]. Similar observations indicating rapid tumor growth as one of the most important characteristics have been published by many researchers [2,3,4]. However, the incidence of uterine sarcoma, including ULMS among patients operated on with rapidly growing uterine tumor compared to patients operated on with non-rapidly growing ULM was not significantly higher—0.7% vs. 0.23%, respectively [13]. In the Nagai et al. study, the rapid growth was seen only in one of fifteen patients with sarcoma, which was less common than in the group of patients with benign lesions [14]. In addition, the definition of rapid growth is unclear, and authors are using different definitions. Rapid growth can be understood as a gain of six weeks or more in gestational size during a year or less [15], therefore taking into account differences between clinical practice and subjective visual assessment, the determination of the probability of malignancy can be confusing. Ghosh et al. observed that ULM growth can range from 18 to 120% per year or its diameter can decrease [16]. The rapidity of ULM growth depends on its location and size. The fastest growth was characteristic of small lesions located intramurally—Mavrelos et al. reported that the median size increase in intramural fibroids was 53.2% and it was higher than the median fibroid volume increase—35.2% [17]. Moreover, it was observed that fibroid growth can be predicted by hormones (FSH, LH) and lipids (LDL, cholesterol) and other blood parameters levels [18]. Unfortunately, it is impossible to compare ULM to ULMS increase, due to a well-known reason; the majority of ULMS is diagnosed postoperatively.

ULMS are diagnosed most often among postmenopausal women and the mean age ranges from 44.6 [19]. to 58.1 [20]. ULM, compared to ULMS, appears at a younger age with a mean age at the time of diagnosis being from 40 [21]. to 51.7 [22]. Furthermore, it may appear in adolescence, although such cases are considered to be rare (less than 1% of ULM) [23]. In the multicenter study by Mayerhofer et al. they reported the association between the age of patients at the time of diagnosis (under 50) and favorable prognosis [24]. In the Brohl et al. study the risk of unexpected sarcoma was dependent on age. The highest sarcoma risk of 10 cases per 1000 was observed for patients aged 75–79 years-old and the lower risk of less than one case per 500 for patients under 30 years of age [25]. Similar conclusions have been drawn by Rodriguez et al. in their study, in which the incidence rate of LMS was the following: 9.8, 10.7 and 33.4 per 10 000 for the ages 25–39, 40–49 and 50–64, respectively [26]. On the other hand, there is some research, including the Qiu Bi et al. [27]. and Kiliç et al. studies [28], in which the age of patients is not considered to have neither positive nor negative impact on the risk. The authors have not noticed any statistically important difference between patients in the uterine sarcoma group and atypical leiomyoma group [27]. Despite the differences between the authors, in the study by Halaska et al. that analyzed the place for morcellation in patients with uterine mass, the age was described as a risk factor of uterine sarcoma, and in older patients morcellation was associated with a higher risk [29]. Köhler’s analysis showed that among all measured symptoms and parameters there are not any specific symptoms differentiating ULMS and ULM. The observations were the following: intermenstrual bleeding was present in 43 per 221 (19.5%) premenopausal women with ULMS compared to 67 per 618 (10.8%) with ULM. In premenopause, dysmenorrhea was more frequent in the ULM group—it was observed in 209 per 612 (36.3%) patients with ULM vs in four per 221 (1.8%) patients with ULMS. Postmenopausal bleeding was more characteristic of the ULMS women—62 per 618 patients (10%), although it has also appeared in 3/618 (0.5%) of ULM women. It is known that all episodes of postmenopausal bleeding should be diagnosed by a specialist, and the tissue from an endometrial biopsy should be evaluated by a pathologist [30]. The presence of symptoms is observed more frequently in the ULM group (303/618 (49%)) versus those in the ULMS group (92/221 (41.6%)), which may be related indirectly to the diagnosis of ULMS with a larger mean tumor diameter [12]. Abnormal uterine bleeding occurring in 23/30 (76.7%) patients with uterine sarcoma and 4/88 (4.5%) patients with atypical leiomyoma was found as statistically significant by Qiu et al. [27]. Taking into consideration all described research there are no specific symptoms for ULMS and the differences regarding its frequency between the two groups were not significant. It is not possible to exclude ULMS in young menstruating women, just as we cannot suspect ULMS in all women of a postmenopausal age. Moreover, since the studies presented above have been performed on small research groups, there are doubts about their credibility.

### 2.2. Specific Markers and Blood Parameters

#### 2.2.1. CA 125

Firstly, analyses of the Carbohydrate Antigen 125 (CA 125) were made, as it is widely used in clinical practice. This marker, associated especially with ovarian cancer, increases not only in this type of cancer, but may also be elevated in 1–2% of healthy people [31], and in 5% of individuals with other conditions, like menstruation [32], pregnancy [33], endometriosis [34], and other inflammatory diseases of the peritoneum.

In ULMS CA 125 was measured for the first time by Duk et al. in 1994. Their results were the following: pre-treatment CA 125 levels were elevated in 12 of 30 (40%) patients with uterine sarcoma and no relationship between CA 125 levels and the histological subtype, including ULMS were observed [35]. The theory about a connection between elevated levels of CA 125 and the presence of ULMS was suggested by Juang et al. [36]. Although, Yilmaz et al. excluded this relationship and did not find it significant based on the analysis of a heterogeneous group of sarcomas [37]. In the research by Skorstad et al., elevated levels of CA 125 (over 35 kU/L) were present in 36/86 women (41.1%) with ULMS. Moreover, they observed that higher levels of CA 125 were correlated with more advanced stage of disease [38]. The newest research by Zhang Fenfen et al. evaluated the usefulness of CA 125 as one of the indicators of ULMS and correlated its levels with lactate dehydrogenase (LDH) and human epididymitis protein 4 (HE4) levels. The specificity, sensitivity, and other characteristics of the scale of Zhang Fenfen et al. are presented in Section 5 Available Scoring Scales [39]. The studies of CA 125 levels in LMS are presented in Table 1. Taking into consideration all the above studies performed on a small research group, the sole verification of CA 125 levels seems not to be useful when it comes to differentiating ULMS and ULM. However, CA 125 may be combined with another indicators, including those presented by Zhang Fenfen et al., HE4 and LDH, blood parameters or/and an image of the lesion in order to increase sensitivity, specificity and accuracy.

**Table 1 cancers-14-01966-t001:** The studies of CA 125 levels in ULMS and ULM group.

The Authors of the Study	The Year of the Study	Research Group	Main Findings
Duk et al. [35].	1994	33 (30 evaluable patients) with sarcoma	12 out of 30 (40% of patients with uterine sarcoma) showed serum CA 125 levels higher than 16 Uml-1. No correlation between serum CA 125 levels and the histologic subtype was found.
Juang et al. [36].	2006	42 patients with ULMS, 84 patients with ULM	Serum CA125 levels were significantly higher in the uterine ULMS group than CA 125 in the ULM group The correlation between CA 125 levels and stage of the ULMS was described.
Yilmaz et al. [37].	2009	26 patients with uterine sarcoma and 2382 patients with ULM	No correlation between CA 125 levels and uterine sarcoma and ULM.
Skorstad et al. [38].	2016	86 patients with ULMS	CA 125 levels higher than 75 kU/L corresponded to more advance stage of disease (FIGO stage IV) compared to the group with CA 125 levels lower than 75 kU/L.
Zhang Fenfen et al. [39].	2021	37 patients with ULMS, 102 patients with degenerated ULM	CA 125 used as one of the LMS indicators may be a promising method to differentiate ULMS and ULM preoperatively.

#### 2.2.2. Lactate Dehydrogenase (LDH)

Lactate dehydrogenase (LDH) is an enzyme catalyzing the conversion of lactate to pyruvate at the end of the glycolytic pathway [40]. Its higher expression has been detected in many cancers as cells, in order to produce more energy, reprogram mitochondrial processes. The only process, in which energy is produced independently from the presence of oxygen is glycolysis, in which LDH is one of the enzymes [41]. Therefore, LDH is connected with tumor growth, maintenance and invasion in many cancers including hepatocellular carcinoma (HCC) [42], breast cancer [43], and others.

The utility of LDH in preoperative ULMS diagnosis was described by Seki et al. in 1992, who observed that serum LDH levels were abnormally elevated in three of seven (42,8%) patients with ULMS [44]. However, LDH is not a specific indicator of ULMS, and it may be present in other types of lesions like ULM or degenerated ULM (DULM) [45]. The other research studies agreed that an elevated level of LDH may help in ULMS and ULM differentiation [46,47]. Additionally, the correlation of LDH levels with other parameters was evaluated. Nishigaya et al. juxtaposed LDH levels with D-Dimer and C-reactive protein (CRP) [48], and Zhang Fenfen et al. with CA 125 and HE4 [39]. (The scales by Zhang Guoruj and Zhang Fenfen are also described in Section 4: Available Scoring Scales). In the study by Zhang Guoruj et al. LDH ≥ 193 U/L was noted as statistically significant and used as a cutoff point in their scoring system [49]. All studies presenting the connection between LDH levels and ULMS are presented in Table 2.

Taking into consideration all the above-mentioned studies, LDH is not seen as a helpful factor in ULM and ULMS differentiation. Its level can be detected higher in other types of malignant tumors, not only in LMS. Therefore, the individual use of LDH seems to have limited advantages.

**Table 2 cancers-14-01966-t002:** The studies of LDH levels in ULMS and ULM group.

The Authors of the Study	The Year of the Study	Research Group	Main Findings
Seki et al. [44].	1992	7 patients with ULMS	Serum LDH levels were elevated in 3 out of 7 (42.8%) patients with ULMS
Goto et al. [45].	2002	10 patients with ULMS and 130 patients with DULM	LDH were elevated in all 10 patients with ULMS and also in 32 patients with DULM
Nagai et al. [47].	2015	15 patients with uterine sarcomas, including 9 ULMS and 48 benign lesions including 42 ULM	Serum LDH levels greater or equal to 279 U/L were observed in 7 out of 15 patients (46.7%) with sarcomas and in none of patients with benign lesions
Kusunoki et al. [46].	2017	15 patients with sarcoma including 6 patients with ULMS and 19 patients with ULM	Levels of LDH in sarcoma group were higher (343 IU/L ± 188) than in LM group (183.1 IU/L ± 44)

#### 2.2.3. Other Tumor Markers: CEA, CA 19-9, CA 15-3

Other tumor markers like carcinoembryonic antigen (CEA), carbohydrate antigen 19-9 (CA 19-9) and carbohydrate antigen 15-3 (CA 15-3) were not used to differentiate ULMS and ULM. There are some studies describing the possible associations of the above-mentioned markers with diagnosis of carcinosarcinoma. As we know, carcinosarcoma has been traditionally considered as a kind of uterine sarcoma, and nowadays it is classified as type II endometrial cancer, however, one of the components of this tumor is mesenchymal [50]. Due to the lack of studies analyzing the markers listed above in uterine sarcomas, we believe that the studies listed below may indicate the potential direction for further research that aims to differentiate ULMS and ULM.

DiSaia et al. observed that CEA was higher in one of three patients with uterine sarcoma, and one of two patients with metastatic uterine sarcoma. However, the histological status is not known, and it has not been compared with patients with ULM [51]. Two other studies evaluated levels of CEA in carcinosarcoma, which were not described as statistically significant [52,53]. To the best of the authors’ knowledge CA 19-9 and CA 15-3 have not been measured, neither in ULM nor in ULMS, and the only available data is their levels of carcinosarcoma.

For now, to the best of the authors’ knowledge, CEA, CA 19-9 and CA 15-3 have not been used for LM and ULMS differentiation, and there is no data suggesting that CEA, CA 19-9, CA 15-3 levels can have any potential importance. Therefore, more research measuring their levels in different histological subtypes of sarcomas and myomas is needed.

#### 2.2.4. Other Parameters

Recently, the blood inflammatory markers and their possible importance in the differentiation of ULMS and ULM have been also analyzed by Suh et al. Their study included the following tests: white blood cell (WBC); platelet; absolute neutrophil (ANC); absolute lymphocyte (ALC) and absolute monocyte (AMC) counts; hemoglobin (Hb) concentration; mean corpuscular volume (MCV); red cell distribution width (RDW); CRP; LDH and CA 125 performed on 79 patients with ULMS and 257 patients with ULM. Among all the performed tests, significant differences were described in the WBC count, ANC, CRP, LDH and NLR [54]. NLR was considered as more useful than serum CA 125 levels by Kim et al. [55]. Moreover, NLR was used as one of the parameters analyzed in the scale of Zhang Guoruj et al. (described in Section 4: Available Scoring Scales), in which the NLR value greater or equal to 2.8 was significant [49]. Moreover, independent meta-analyses of Liu et al. on soft tissue sarcoma and Jiang et al. on bone sarcoma have shown that elevated NLR was highly correlated with worse overall survival rates [56,57]. Taking into consideration the fact that the results of research considering blood inflammatory markers, especially NLR as biomarkers for the differentiation of uterine ULMS and ULM are promising, more research should be conducted, and perhaps, afterwards some new scales should be conceived.

## 3. Imaging Methods

### 3.1. Ultrasound

Ultrasound, including transabdominal ultrasound and transvaginal ultrasound, is usually the first imaging method used for detection and initial classification of uterine masses. In the ultrasound examination, leiomyomas are described as solid, well-defined, concentric masses with heterogeneous echogenicity, depending on the presence of calcifications or/and the amount of fibrous tissue [58]. ULMS are imaged quite similarly—as large oval-shaped inhomogeneous masses distorting the uterine architecture with areas of necrosis and hemorrhages [59,60].

The first more detailed analysis conducted to find parameters differentiating ULMS and ULM was published in 1997 and was based on intra-tumoral blood flow velocity waveforms with Doppler ultrasonography. In this study, Hata et al. calculated the resistance index (RI) and peak systolic velocity (PSV), among which RI turned out to be negligible. Nevertheless, PSV, the other measured parameter, was significantly higher in the uterine sarcoma group than in the uterine leiomyoma group. Assuming a cut-off value for PSV of 41.0 cm/s, the detection rate for uterine sarcoma was 80.0%, and the negative predictive value was 97.6%. This indicator was defined to be very promising, although only five patients with ULMS were taken into consideration in the above-mentioned study [61]. The lack of utility of RI was described by Szabo et al., who admitted that RI is not effective for the preoperative differential diagnosis of ULMS. They also reported that the increase in PSV and the reduction in non-specific RI was found in 14 out of 117 (12%) ULM cases, which presented characteristics of ULMS, such as large size and/or necrotic, degenerative and inflammatory changes [62]. Aviram et al. focused on the differentiation of uterine sarcomas of different histologic types compared with ULM by contributing to gray-scale sonography and Doppler flow. They reported that as far as the distinction of ULM and malignant mixed mesodermal tumors, taking into account the mean resistance index in arterioles, is possible, it is not possible to differentiate ULM and ULMS [63]. Those conclusions were not consistent in the results of Exacoustos et al., who also sought to evaluate the role of gray-scale and color Doppler sonography to distinguish ULMS from ULM. Their results showed some differences between ULM and ULMS, like diameter > or = 8 cm (87.5% of ULMS), degenerative cystic changes (50% of ULMS) and increased peripheral and central vascularity (87.5% of ULMS), what, according to the authors can identify suspicious uterine smooth muscle tumors [64]. In the literature there is also some evidence that in order to differentiate ULMS, adenomyosis and ULM, the evaluation of the endo/myometrial junctional zone may be significant, especially the vascular pattern between these two types of tissue, as was suggested by Fascilla et al. [65]. Russo et al. have also suggested that a type of vascularization (circumferential and intralesional) can help in benign and malignant lesion differentiation [66]. All of the studies quoted in the ultrasound part of our review took into consideration a small group of patients, the largest of which had a number of 13 and did not help in ULM and ULMS differentiation. Yet, in 2019, to the best of the authors’ knowledge, the largest study describing ultrasound findings in uterine sarcomas was published. The data of 195 patients with uterine sarcoma, including 116 patients with ULMS, was analyzed and the results were the following: the type of tumor was described as solid in 85 out of 116 (73.3%), the echogenicity of solid tumor was inhomogeneous in the majority of patients: 94/116 (81%), cystic areas were present in 54/116 patients (46.6%), calcifications were present in 18/116 patients (15.6%), tumor border was regular in 55/116 patients (47.4%) and irregular in 61/116 patients (52.6%). A cooked appearance, understood as a lack of structure of the solid tissue of the tumor, was observed in 317/1284 patients (24.7%) in the study [67].

Although it is difficult to imagine our daily living without ultrasound and the majority of gynecological diseases are being diagnosed thanks to imaging, the usage of ultrasound in the differentiation of ULM and ULMS seems to have been described as limited. Due to ULMS being a rare neoplasm and lack of clear indication, ULMS may be overlooked in imaging even by an experienced specialist. In addition, all published studies on USS are retrospective and not many patients are included. The largest study analyzing 116 patients with ULMS did not compare the image of ULMS with ULM, therefore did not give an instruction on how to distinguish these two types of lesions. Considering all the limitations listed above, reliable differentiation between uterine ULM and ULMS based only on USS seems to be impossible so far.

### 3.2. Computed Tomography (CT)

CT plays a limited role in the preoperative diagnosis, staging and differentiation of uterine mass. In CT after contrast administration, sarcomas are described as a large, slightly heterogeneous mass replacing the uterus, with obvious enhancement and an early phase of peripheral necrosis, which is similar to the ultrasound view [68]. The predominance of CT is based on the possibility of it showing a detailed view of calcifications, however, the usefulness of measured calcifications is doubtful [69,70]. Lee et al. agreed that calcifications may help in leiomyomatous tumor diagnosis, nevertheless, differentiation of ULMS and ULM is not described as being any easier thanks to this measurement. In the study, they noted calcifications in four of the ten ULMS, and one of the two LM. It is important to add that they focused on tumors of the colon and the rectum, not the uterus, thus some differences depending on the tissue origin are possible [71]. Unfortunately, to the best of the authors’ knowledge, no research evaluating the frequency of calcifications in the uterus with ULM or ULMS was published. CT, as one of the most frequent and one of the cheapest imaging methods, is useful in determining the stage of the disease [72], or in the evaluation of complications after hysterectomy [73], but there is no data proving its value in ULMS and ULM differentiation.

### 3.3. Magnetic Resonance Imaging (MRI)

The radiologists agreed that it is quite difficult to differentiate ULM from ULMS based on Grayscale/color Doppler ultrasound and CT. One of the most important features of MRI is better morphologic information on soft tissue intensity, therefore it is possible to classify a uterine mass as either a benign or malignant tumor with more precision. In their study, Hélage et al. observed some differences in malignant lesions on the MRI image and divided them in two types: type 1: intracavitary (for example—endometrial stromal sarcomas); and type 2: intramyometrial with ULMS as an example. In the ULMS group, the characteristics are irregular margins and tumor heterogeneity [74]. Moreover, a feature distinctive of sarcoma, but not a reliable indicator of malignancy, is high signal intensity (SI) [75]. On T2-weighted images (T2WI) a hyper-intense signal suggests high cellularity or high vascularity of uterine sarcoma. A high signal can be also described on T1-weighted images (T1WI) as an indication of intratumoral hemorrhage and coagulative necrosis, which also suggests sarcoma of the uterus. (13) In comparison, ULM manifests as a mass presenting low SI as well on T1WI, as on T2WI. There are some studies suggesting high signal on T1WI is more relevant to ULMS and ULM distinguishment [76,77]. However, there are also studies considering high intensity on T2WI to be more typical for malignant lesions [78], or, regarding ULMS and ULM differentiation, some typical characteristics are visible not only in one, but in two sequences [79].

Even if the authors found one sequence more important to differentiate ULMS and ULM, the characteristics are not visible in all cases. For example, in the study by Ando et al., hyper-intense areas on T1WI were not typical for all ULMS, but for approximately 80% of them [76]. The fact that not only high SI is an important indicator of malignancy was suggested also by Kido et al. In their opinion, the detection of hemorrhage or coagulative necrosis, which are described as areas with high SI on T1WI and heterogeneous areas on T2WI, are the most important factors in differentiating ULM and ULMS [80]. In addition, in the Lakhman et al. study, two independent readers compared characteristics of ULMS with atypical ULM. They have indicated that nodular borders, T2 dark areas and central unenhanced areas are more characteristic for ULMS, [81] as well as the size of the tumor—ULMS diameter measured by MRI was observed to be greater compared to ULM [11,82].

Diffusion-weighted imaging (DWI), which reveals tissue characteristics on the base of the diffusion motion of water molecules, seems especially useful in ULMS and ULM distinguishing. Apparent diffusion coefficient (ADC) is two or three times lower in the tissue compared to water and is lower in malignant tumors than in normal tissue. Malignant tumors, including ULMS, present high SI on DWI with significantly lower ADC than in benign lesions, as has been observed by Takeuchi [83]. The role of ADC in ULM and ULMS was also investigated by Tasaki et al. in 2015. They observed that the mean ADC was significantly lower for ULMS than for ULM, while the mean cell density was significantly higher for ULMS than for ULM [84]. Hai Ming Li observed that ADC is helpful not only in ULM and ULMS differentiation, but that this indicator is also different in ULMS and degenerated leiomyoma (DLM), which are the most difficult to distinguish preoperatively [85]. DWI combined with T2WI significantly improves the possibilities to distinguish uterine sarcomas from benign leiomyomas, as has been shown by Namimoto et al. The authors compared the sensitivity and specificity of DWI alone and DWI combined with T2WI on the 103 patients with 103 myometrial masses, including eight ULMS and 95 ULM. The indicators used in the study were tumor-myometrium contrast ratio (TCR) counted as SItumour–SImyometrium)/SImyometrium and ADC. DWI combined with T2WI demonstrated the highest sensitivity and the highest specificity, both estimated at 100%. Even though the study is very promising, the authors cited many limitations, such as the small number of uterine sarcoma cases or the lack of classification of leiomyomas into histopatological subtypes. However, the obtained results fully justify further research evaluating this method [86].

The algorithms based on MRI are most valuable for the future and crucial for clinical work development. The first ones, submitted in 2013 by Thomassin-Naggara et al., presented a model differentiating malignant from benign myometrial tumors of the uterus. The MRI criteria predicting malignancy were the following: high b 1000 signal intensity; intermediate T2-weighted signal intensity; mean ADC; patient’s age; intra-tumoral hemorrhage; endometrial thickening; T2-weighted signal heterogeneity; menopausal status; enhancement heterogeneity; and the non-myometrial origin on MRI, as the most significant features of malignancy. Despite the promising result of their study, the authors included not only ULMS, but also malignant mesenchymal uterine tumors, and therefore, it has limited possibilities with regard to the differentiation between ULMS and ULM. (The clinical parameters, specificity, sensitivity and limitations of the study are presented in Section 4: Available Scoring Scales.) [87]. The scale prepared by Jagannathan marked seven out of fifteen features as the most important: heterogeneity on T2w images, hyper-intensity of the solid component on T2w images, intra-tumoral hemorrhage, heterogeneous enhancement, enhancing finger-like projections, ill-defined border with myometrium on post-contrast T1w images, and central non-enhancing necrosis [88]. The next one, proposed by Wahab CA et al., uses four MRI criteria: enlarged lymph nodes or peritoneal implants; presence of the focal region or global low SI in T2WI; visual analysis of DWI signal intensity as low, intermediate or high in reference to myometrium and endometrium and/or lymph nodes; and the last one—ADC value. The sensitivity of a less experienced reader was 83% and the specificity was 97% after the first read, which in our opinion accomplished a very high notification, and therefore, the scale could be useful for other specialists, even those with less experience [89]. Taking into account the studies presented above, ADC seems to be the most important indicator of ULMS, yet, this opinion has been disproved by Kaganov et al., who have observed that ADC values were not defined to have a significant statistical correlation with ULMS and benign uterine lesions. The authors noticed this may be connected with the variable and complex structure of the lesion, including higher ADC values residing not only in solid parts of sarcomas, but also in degenerative ULM [90]. To the best of the authors’ knowledge, this is the only study, which denies ADC having meaning with regard to ULMS and ULM distinguishing.

The studies analyzing T1 and T2 and DWI sequences are presented in Table 3.

Unfortunately, all the above-mentioned studies describing MRI findings are retrospective and based on a small number of patients in the research group, especially LMS cases. Many authors did not divide research groups to the subgroups based on histological types of the lesions. We should not forget that, despite promising results, MRI is not available in all centers where surgery on ULM is performed. As a consequence, we cannot perform MRI on all patients. Among all the accessible methods, DWI, and especially DWI combined with T2WI with ADC evaluation, seems to be the most effective and precise method of distinguishing ULM and ULMS.

**Table 3 cancers-14-01966-t003:** MRI in preoperative diagnosis of uterine mass.

Sequences	The Authors of the Study	The Year of the Study	The Number of Patients	Main Results
T1 and T2	Tanaka et al. [79].	2004	24 women (including nine ULMS and 12 cases, in which gynecologists suspected leiomyosarcomas)	The presence of high SI on T2WI and any small high-signal areas on T1WI with unenhanced regions were characteristic for ULMS.
	Nagai et al. [14].	2014	63 women (including nine ULMS and 42 ULM)	MRI findings were used as one of the components in the PREoperative Sarcoma Score (PRESS).
	Ando et al. [76].	2018	509 women with 1137 uterine smooth muscle tumors (including 14 ULMS and 1118 ULM)	Hyper-intense areas on T1WI of ULM were characterized by more homogeneity, better demarcation, smaller occupying rate and higher signal intensity than hyper-intense areas on T1WI of ULMS.
	Lakhman et al. [81].	2016	41 women (22 atypical ULM, 19 ULMS)	Three or more qualitative MR features (nodular borders, hemorrhage, “T2 dark” area(s) and central unenhanced area (s)) were helpful in ULM and ULMS differentiation. ULMS on T2WI was characterized by intermediate signal and irregular margins.
	Hélage et al. [74].	2021	50 women (including 19 ULMS)	ULMS on T2WI was characterized by intermediate signal and irregular margins.
	Jagannathan et al. [88].	2021	44 women (including 19 with ULMS and 25 with ULM)	Seven out of 15 MR imaging features were found to be useful to distinguish ULMS and ULM.
DWI	Takeuchi et al. [83].	2009	34 women (including one ULMS and 27 ULM)	High SI on DWI with low ADC was more characteristic for malignant than for benign masses.
	Tasaki et al. [84].	2015	144 women with 168 lesions (including six ULMS and 159 ULM)	ADC is helpful in benign and malignant tumors with high SI on T2WI and DWI differentiation.
	Li et al. [85].	2017	42 women (including 16 ULMS and 26 degenerated ULM)	The mean ADC value in ULMS was significantly lower compared to degenerated ULM.
	Hélage et al. [74].	2021	50 women (including 19 ULMS)	ADC values lesser or equal 0.86 × 10^−3^ mm^2^/s were suggestive of malignancy

### 3.4. Fluorine-18-Fluorodeoxyglucose Positron Emission Tomography (18)F-FDG (PET)

One of the most characteristic biochemical markers of cellular transformation in malignant tumors is an accelerated rate of glucose transport. It is associated with an increase in glucose transporter protein and also with an elevation of glucose transporter messenger RNA [91]. As 18F-FDG uptake was expected to be greater in ULMS, PET was analyzed as one of the imaging methods differentiating ULM and ULMS.

The diagnostic accuracy of PET for LMS diagnosis was estimated at 73% by Nagamatsu et al. [92]. The study of Umesaki et al. preoperatively compared the use of three imaging methods: PET, MR and US in uterine mass classification and, as a result, it showed the highest efficacy of PET (100% of efficacy) compared to MR (80% of efficacy) and US (40% of efficacy). This study was performed on a small research group counting only five patients, including three patients with ULMS with no control group [93]. Park et al. have not observed any differences between MRI and PET/CT in sensitivity, specificity, accuracy, or positive and negative predictive values in detecting primary uterine lesions [94]. Lee et al., in their retrospective study, observed that preoperative tumor heterogeneity index obtained using MTV linear regression slop may be a very helpful prognostic marker in ULMS and ULM differentiation [95]. It was also noticed that PET is better than MRI in detecting distant metastases. In the Sung et al. study six out of seven cases of uterine sarcomas (85.7%) were diagnosed based on PET images, fewer than based on CT scans (7/7—100%). The authors of the study noticed that PET showed a better detection of extrapelvic metastases compared to CT scans; seven metastatic sites were described in PET and three observed in CT. Similarly, many authors perceived that PET utility is especially significant in recurrent uterine ULMS, in the follow-up of patients with sarcomas [96,97,98], nevertheless, no meaningful difference was noticed for ULM and ULMS differentiation.

To summarize, we agree with the results of the systematic review and meta-analysis performed by Sadeghi et al. suggesting that PET appears to be an accurate method for detection and localization of recurrence in patients with uterine sarcoma. The authors emphasized that there is a limited access to studies analyzing the role of PET in primary staging of uterine sarcoma, therefore no conclusions in this area have been drawn [99]. Taking into consideration all the studies mentioned above it seems that PET may be a better method to diagnose uterine mass and to detect metastases in the follow-up of patients. However, the cost, low availability and the difficulties in performing the test are some of the examples of obstructions of its daily use.

### 3.5. Radiomics and Artificial Intelligence (AI)

Nowadays, many authors are analyzing if the use of a machine learning model can be helpful in tumor differentiation, especially regarding the rare ones. Therefore, the utility of radiomics was also described in ULM and ULMS diagnosing. (Table 4).

The potential usefulness of perfusion weighted MRI (PWI) was described by Malek et al. in their machine learning model. In order to find parameters, which could help in ULM and ULMS differentiation, an experienced radiologist manually outlined two regions of interest—one represented the entire tumor, and the second—the area of the lesion with the most marked contrast enhancement. They did not find one specific parameter, which would be significantly different between benign and malignant lesions, however, when 21 extracted features were used, sensitivity and specificity were 100% and 90%, respectively [100]. In order to differentiate atypical ULM and ULMS, the study of Xie et al. analyzed three different volumes of interests (VOIs) using radiomics. They divided regions in ADC into the following groups: (1) tumor; (2) tumor and small piece of surrounded tissue; and (3) whole uterus. The best diagnostic performance of a VOI radiomic model was one that covered the whole uterus including the highest AUC, sensitivity, specificity and accuracy [101]. The results of some studies suggest that the efficacy of the radiomics model is comparable to the efficacy of experienced radiologists [102], however, some of the authors observed that the use of radiomics outperformed the efficacy of radiologists’ diagnosis [103]. The interesting machine learning model with the usage of radiomics features in hyperintense T2WI MRI scans and clinical information has been presented by Wang et al. Clinical characteristics, like symptoms, age and menopausal status were compared to the MRI image evaluated by two blinded, experienced radiologists. It was observed that the results of both clinical and radiomics features obtained by computer-aided diagnosis (CAD) had higher performance than the results achieved by two professional radiologists. In addition, the limitations of the study could impact on the results of the study. Among others, the most important seems to be the fact that the authors qualified stromal tumors of uncertain malignant potential to the malignant group, although in the majority of studies, they are analyzed separately [104].

Big potential relates to the use of radiomics in ultrasound, which was described in an ADMIRAL trial made by Chiappa et al. In the study, 20 women with histological diagnosis of sarcoma and 50 with myomas were classified, 390 radiomic IBSI-compliant features were extracted and 308 radiomic features were found to be stable. The best classification system showed an accuracy of 0.85, sensitivity of 0.80, specificity of 0.87, AUC of 0.86. The study also had some limitations as a low number of cases had enrolled and, as such, in the future, research based on a greater number of patients should be performed [105].

Our observations are consistent with the conclusion of a systematic review analyzing the potential use of radiomics and artificial intelligence by Ravegnini et al. In their opinion, there is not enough evidence indicating the benefits of the use of radiomics to diagnose uterine sarcomas [106]. We should not forget about limitations of the study: the small number of patients (the greatest number of patients in the study of Nakagawa et al. was 80), the small number of studies carried out, and the retrospectivity of the studies. However, the development of artificial intelligence seems to be one of the main goals of our current age, and even if the results of the above-mentioned studies are not yet satisfying, we believe that in future it could be one of the best methods of LM and LMS differentiation.

**Table 4 cancers-14-01966-t004:** Radiomics in uterine masses differentiation.

Authors of the Study	Year	Diagnostic Method	The Number of Patients	Main Findings
Malek et al. [100]	2019	MRI	42 patients with 60 uterine lesions: 10 uterine sarcomas and 50 ULM	They extracted 21 radiomics features achieving the sensitivity and the specificity: 100% and 90%, respectively
Xie et al. [101]	2019	MRI	78 patients with 29 sarcomas and 49 ULM	They analyzed three different volumes of interests (VOIs.): (1) tumor; (2) tumor and small piece of surrounded tissue; and (3) whole uterus. The best diagnostic performance of the VOI radiomic model was one that covered the whole uterus.
Wang et al. [104]	2021	MRI	134 patients including 81 with LM and 53 malignant uterine mesenchymal tumors	They used three clinical parameters: the age, menopausal status, and symptoms and seven radiomic features. The highest AUC value was the highest when the radiomics model was combined with a clinical model to 0.91.
Chiappa et al. [105]	2021	Ultrasound	70 patients with uterine mesenchymal lesions: 20 with sarcoma and 50 ULM	390 radiomic IBSI-compliant features were extracted and 308 radiomic features were found to be stable. The best classification system showed an accuracy of 0.85, sensitivity of 0.80, specificity of 0.87, AUC of 0.86.

## 4. Molecular Features

### 4.1. The Usefulness of Endometrial Biopsy

The histopathologic evaluation of surgical specimens remains the gold standard in assessing whether a myometrial lesion is benign or malignant [2]. Nowadays, making a diagnosis preoperatively is especially important, as in the majority of cases it decides which surgical method would be appropriate. In the cases of fibroids, in which morcellation is used, it may change the prognosis completely, when it turns out to be unexpected sarcoma. Therefore, the idea of biopsy usage in the ULM and ULMS preoperative differentiation occurred. European Society of Gynecological Oncology emphasizes that preoperative endometrial biopsy should be mandatory, before surgeries associated with using morcellation except other malignancies, especially endometrial carcinoma or carcinosarcoma [3,10,11]. This makes the methods used in the diagnostic process even more important [10].

Effectiveness of biopsy and curettage in identification or suspicion for malignancy varies from 35.5% to 86%, and in the identification of leiomyosarcoma specifically from 51.5% to 64% [38,107,108] (the detailed results of studies are presented in Table 5). It is worth mentioning that there is no statistically significant difference between dilatation and curettage and endometrial biopsy (pipelle) with regard to effectiveness, and that is comparable for both of them—66% vs. 61% in the study of Bansal et al. [107], and 56% vs. 48.8% in the study by Hinchcliff et al. [108]. Nevertheless, some authors point to the superiority of pipelle biopsy due to the lower invasiveness of the procedure compared to dilation curettage [107].

**Table 5 cancers-14-01966-t005:** The effectiveness of endometrial sampling in ULMS diagnosing.

Study	Year of the Study	Preoperative Endometrial Sampling, Which Identified Malignancy or Suspicion of Malignancy	Preoperative Endometrial Sampling, Which Identified Leiomyosarcoma Specifically
Bansal et al. [107].	2008	86% (46/72) *	64% (46/72) *
Hinchcliff et al. [108].	2016	35.5% (24/68)	51.5% (35/68)
Skorstad et al. [38].	2016	38.7% (55/142)	-

* all types of uterine sarcomas.

The biopsy in ULM and ULMS cases has many limitations. The first of them is connected with the location of those lesions—ULM and ULMS are derived from the deep muscular myometrial layer of the uterus, and are less accessible rather than the superficial endometrium, which in the majority of cases does not show any abnormalities [2,107,109]. In addition, in the histopathological assessment according to the commonly used classification (degree of malignancy of common-type smooth muscle tumors) proposed by Bell et al. in 1994, the evaluation of the sampling based on the three factors: mitotic index; the degree of cytological atypia; and the presence or absence of coagulative tumor cell necrosis; should be performed in the most advanced areas [110]. Unfortunately, these areas are often unavailable in endometrial biopsy sampling. Another possible critical drawback during the procedure is the spreading of tumor cells or infection, however, taking into consideration the frequency of those complications, it is not seen as a crucial problem [111].

In the study by Hinchcliff et al. (148 patients with ULMS, including 68, who underwent preoperative endometrial sampling) it was observed that patients with postmenopausal bleeding in comparison with these with other symptoms (premenopausal bleeding, bulk symptoms, pain) were significantly more likely to be qualified for preoperative tissue sampling. It may be associated with the fact that postmenopausal bleeding is rarer than premenopausal bleeding, thus it may be worrisome and lead people to undergo diagnostic procedures. Moreover, it turned out that higher sensitivity in detecting malignancies was found in patients with postmenopausal bleeding than in patients with other previously mentioned symptoms (72.7% vs. 32.3%) [108].

What is more, in a study by Kho et al. performed on a group of 79 patients with ULMS it, was noticed that higher chances to detect ULMS preoperatively in these women, who had undergone endometrial sampling supplied with hysteroscopy. ULMS was diagnosed in 40 of 60 (66.7%) patients due to the additional usage of hysteroscopy compared to half as many in the group undergoing only endometrial sampling (6 of 19 patients, 31.6%). Moreover, it was observed that it was less likely to detect LMS in endometrial sampling in cases of sarcoma at an early stage of disease, and when the size of the tumor was greater than 11 cm [112].

Nevertheless, considering the low cost of both methods (biopsy, curettage) and the low risk of complications they may cause, their use in preoperative diagnostics seems justified. Even its utility in ULMS is disputable, its application is helpful in endometrial cancer or carcinosarcoma diagnosis. Moreover, biopsy of endometrium was used by Lawlor et al. in their scale (described in Section 4: Available Scoring Scales), in which a higher risk of ULMS was connected with endometrial biopsy results of cellular atypia or neoplasia [113]. Taking into consideration the Kho et al. study, some hopes can be connected with endometrial sampling with hysteroscopy [112], although in our opinion it is expensive and impossible to implement in all patients with leiomyomas.

### 4.2. The Usefulness of Needle Biopsy

The next doubt is the effectiveness of needle biopsy in ULMS and ULM differentiation. In the study by Yoshida et al., 475 patients underwent transcervical needle biopsy, and among them eight ULMS cases were found. By analyzing biopsy material, they prepared a scoring system, in which not only histopathological, but also immunohistochemical features were studied. Histopathological evaluation included the degree of cytologic atypia, mitotic index and coagulative necrosis (CTCN). Moreover, Ki-67, a proliferation marker for human tumor cells, were also examined, as well as CD34 expression in cases of suspected CTCN. The CD34 is a protein, the expression of which is observed in early hematopoietic and vascular-associated tissue, therefore, it was used in order to check the presence of coagulative tumor cell necrosis. Cases which have two or more positive factors (six or more points in the scale of 19 points) were classified as LMS. The positive predictive value, when the cut-off score was six points, was 100%, therefore it may become a very useful scoring system for cases that included a needle biopsy before surgery. However, as we know, not all patients undergo a biopsy before surgery, and the study was retrospective. Therefore, it should be evaluated in a prospective study [114].

Telomerase activity also seems to be another promising factor, which can be helpful to distinguish uterine sarcomas and leiomyomas preoperatively. In the study of Tsujimura et al., 62 patients with high risk of sarcomas underwent transcervical ultrasound-guided biopsy. Higher activity levels of telomerase were detected (measured by telomeric repeat amplification protocol (TRAP)) in sarcomas cases (22 to 102 units), in comparison to benign samples (11–18 units). The sensitivity and specificity were noted as 86% and 100%, respectively. It is worth adding that it requires complex preparation; the sample must be kept in −80 Celsius degrees until measurement to avoid telomerase deactivation [115], which make the process more difficult and not practical for wide application.

The utility of needle biopsy is not well described in ULMS and ULM differentiation. In our opinion it is connected to the fact that its application is more complicated than others methods, and the price is higher. Nevertheless, the above-mentioned studies suggest that its usage and measurement of CD34 or activity of telomerase in the tissue may have potential, although more research is needed to confirm the thesis.

### 4.3. Circulating Biomarkers

Another new possibility, which can be tested in the blood, is growth differentiation factor 15 (GDF-15) and models taking into account specific microRNA (miRNA). GDF-15, a divergent member of the TGF-β superfamily, is connected with the evolution of cancer, however, its activity depends on the cellular state and environment. It was shown that abnormal expression in cancers intensifies proliferation, invasion, metastasis and reduces response to therapy [116]. In the study by Trovik et al. the differences of GDF-15 levels between sarcoma group (19 patients) and myomas group (50 patients) were observed. An especially high GDF-15 median level of 1397 ng/L, compared to the uterine sarcoma group (943 ng/L) and ULM group (647 ng/L), suggests that it may be a useful ULMS marker for preoperative differentiation of benign and malignant uterine lesions [59]. Yokoi et al. in their study wanted to check if miRNA may be one of the ULMS biomarkers, which could simplify the diagnosis of this type of tumor. From 7 miRNA with the highest cross-validation score, they selected two: miR-1246 and miR-191-5p, which had the highest AUC rates. The authors suggest that miRNA could help in ULMS and ULM preoperative differentiation, although future validation and additional optimization of the presently identified model is required [117]. All of the alternative methods, including immunochemistry, markers in the tissue, and in the blood, are presented in Table 6.

**Table 6 cancers-14-01966-t006:** The utility of telomerase activity, Ki-67, CD34, GDF15, miRNA in ULMS and ULM differentiation.

Name of the Marker, Receptor	Sample Source	Study	Year of the Study	Researched Group	Main Findings
Telomerase activity	Tissue	Tsujimura, et al. [115]	2002	62 (including 6 ULMS, 53 ULM)	The telomerase activity is significantly higher in uterine sarcoma than in ULM. The tissue should be observed histopathologically to determine whether necrotic tissue is present, cause then telomerase may be negative
Ki-67, CD34	Tissue	Yoshida et al. [114]	2009	475 (including 8 ULMS)	Ki-67 and CD34 were used as one of the markers in their scoring scale. Ki-67 was performed in all patients and CD34 expression was added if coagulative tumor cell necrosis was observed. All ULMS samples had Ki-67 labeling index was 15% or more.
GDF-15	Blood	Trovik, et al. [59]	2014	109 (including 13 ULMS, 50 ULM)	The median circulating GDF-15 concentration was elevated in the uterine sarcoma group and was (943 ng/L) in contrast to the myoma uteri group, where it was (647 ng/L). Its level was significantly higher in patients with metastatic disease, with large tumor diameter, and with leiomyosarcomas as compared with other histological types.
miRNA	Blood	Yokoi et al. [117]	2019	29 patients (including 6 ULMS and 18 ULM)	7 types of miRNAs (miR-4430, miR-6511b-5p, miR-451a, miR-448 5-5p, miR-4635, miR-1246 and miR-191-5p) were selected as potential markers for the diagnosis of leiomyosarcoma. Optimal prediction model (miR-191-5p and miR-1246) was proposed, and its detection of accuracy is similar to that of MRI

When more invasive procedures are considered in ULM and ULMS differentiation, we should deliberate not only on its effectiveness, but also on its availability, price, or difficulties in performing them. It seems that endometrial biopsy is the most accessible option, which, in addition, is recommended by the European Society of Gynecological Oncology. However, it is doubtful whether its efficacy of nearly 60% will influence further proceedings, even if the results indicate ULMS. On the other hand, it may be helpful in endometrial cancer exception or as a support in the scoring scales, like in Lawlor et al.’s scale. More hopes are connected with needle biopsy and CD34 or telomerase activity evaluation, even if the doubts are connected to the availability and difficulties in performing them. To the best of the authors’ knowledge, the studies of telomerase activity and CD34 in sarcomas were finished more than 10 years ago and have not been continued, therefore we do not think they can change ULMS and ULM differentiations in the future. In our opinion, microRNA or GDF-15 are among the markers which could reduce difficulties in uterine mass diagnosis. It relates to the fact that access to genetic research is becoming more and more easy and cheaper to evaluate. GDF-15 can be tested in blood, the process does not require specific conditions, and the differences in its levels in the benign and malignant groups were significant. For now, all studies were retrospective and performed on small number of patients, therefore more research, especially prospective, is needed.

## 5. Available Scoring Scales

Numerous scoring scales have been elaborated in the aim of facilitating preoperative ULM and ULMS distinguishing. Some of them are grounded on laboratory tests, others rely on imaging methods or/and molecular aspects. Scales based on parameters from different categories seem to be the most innovative.

In the scale of Nagai et al., named as the PREoperative Sarcoma Score (PRESS), they evaluated the utility of clinical findings, blood tests, imaging studies (USS and MRI) and endometrial cytology of 63 suspected uterine sarcoma cases. From all parameters, age, LDH, MRI and endometrial cytology findings were the most important predictors for sarcoma. In their seven-point system, two points were given in the cases of age ≥ 49, when the serum LDH values were ≥279 and where cytological findings were present. Furthermore, one point was for positive MRI finding. The authors used three points as a cut-off score—the patients with a score of three or higher should undergo surgery and those with two points or less could avoid it. In their study, three of fifteen patients with uterine sarcomas had a score of two points. Nevertheless, 12 patients were classified properly, therefore sensitivity and specificity were 80% and 85.4%, respectively [13]. Preoperative differentiation of ULM and ULMS in the Köhler et al. scale has been determined using clinical parameters such as bleeding symptoms: intermenstrual bleeding; hypermenorrhea; dysmenorrhea; postmenstrual bleeding; and imaging features such as suspicious sonography and the tumor diameter. This study included the greatest number of patients: 293 with ULMS and 826 with ULM, which allowed them to create a reliable classification score. The most important authors’ conclusion concerned the possibility of predicting ULMS and the implementation of subsequent diagnostic techniques including endometrial biopsy, color Doppler sonography, LDH levels measurement and transcervical biopsy, in the cases which were not classified, neither as ULM nor as ULMS group. The study had some limitations: retrospective collection of data and the differences in rapid growth diagnosis [12]. Thomasin-Naggara et al. also based their work on the age of the patient, menopausal status and MRI parameters including high b1000 signal intensity, intermediate T2-weighted signal intensity, mean, intra-tumoral haemorrhage, endometrial thickening, T2-weighted signal heterogeneity, heterogeneous enhancement and non-myometrial origin on MRI. Among all the parameters, the authors have noticed that the T2 signal, high b1000 signal and ADC measurement are the most relevant, although higher age and menopausal status of the patient have been observed in patients with malignant masses, however the MRI parameters seem to be more important. The aim of the study was to differentiate benign and malignant lesions of the uterus, therefore the results could have been different if only ULM and ULMS had been taken into analysis [87].

Nishigaya et al. evaluated the diagnostic value of the combination of serum concentration of LDH, D-dimer and CRP. Positive rates of LDH have been observed in 66.7% of patients in the ULMS group, in 14.3% in the presumed malignancy group and in none of the ULM patients. As far as the second measured parameter is concerned, D-dimer levels were elevated in 83.3% of the ULM + S cases, 17.9% of the presumed malignancy cases and in 5% of the ULM cases. CRP rates were positive in 64.5% of the ULMS patients, 10.7% of the presumed malignancy patients and 2.9% of the ULM patients. The specificity was measured as 100% with a usage of a combination of LDH and D-Dimer, as well as a combination of all the parameters. The sensitivity of the scale was, importantly, lower—35.3%. The authors emphasized that not all the parameters were available in every patient, it was a retrospective study, and the number of patients was limited [48].

Similar to Nishigaya et al., Zhang Fenfeng et al. used LDH levels since their first test and compared its levels with serum concentrations of CA125 and HE4. The sensitivity was 68.4% and the specificity was 95.1% In their study a combination of the above-mentioned markers had better diagnostic efficacy than any pairwise combination of markers or by any marker used alone (CA125: the sensitivity—36.8%, the specificity—90.2% with a cut-off value of 30.85 U/mL; LDH: the sensitivity: 72.7%, the specificity: 87.3%; HE4: the sensitivity—63.2%, the specificity—75.5%). The authors reported certain downsides, such as retrospectivity of the study and the small number of patients as well [39]. One of the biggest advantages of Zhang Fenfen et al.’s research is the fact that all tests are available and cheap, and therefore it is possible to perform them on the majority of patients.

Undoubtedly, imaging methods increase the effectiveness of laboratory tests. It has been observed by Goto et al., who compared the specificity, the sensitivity, the positive predictive value, the negative predictive value and the diagnostic accuracy in patients with ULMS and ULM. The sensitivity was of 100% for both parameters: MRI and LDH alone, as well as combined. Joining LDH and MRI increased the specificity from 87.7% for LDH levels and 96.9% for MRI to 99.2% [45].

One of the most complex and accurate scoring systems was that presented by Zhang Guoruj et al. With the aim of distinguishing ULM from ULMS, Zhang Guoruj et al. based their study on the following parameters: age ≥ 40 years old, tumor size ≥ 7 cm, neutrophil-to-lymphocyte ratio (NLR) ≥ 2.8, number of platelet ≥ 298 × 109/L and lactate dehydrogenase (LDH) ≥ 193 U/L. The total score of the scale was seven points, from which two points were given for tumor size ≥ 7 cm and LDH ≥ 193 U/L, and other parameters such as age, neutrophil-to-lymphocyte ratio, and the number of platelets were assigned one point. Score ≥ 4 points was an useful predictor in the differentiation of ULMS from ULM. This study was performed on the greatest number of LMS patients—45, however, the study had some limitations. Previously, it was a single center retrospective study, therefore, multicenter, prospective studies with greater numbers of patients are needed [49].

One of the newest methods, radiomics, has been also used with the purpose of distinguishing ULM and ULMS preoperatively. As it was presented earlier in the MRI section, Wang et al. created a machine learning model, which had a better AUC (area under curve) (0.91) value in comparison to the radiomics model (0.76) and the clinical model (0.79). In addition, the result of the clinical-radiomics model was more accurate than the evaluation of lesions made by two radiologists [104]. Therefore, this method appears particularly promising.

All the aforementioned scales are presented in Table 7.

**Table 7 cancers-14-01966-t007:** The analysis of the available scoring scales (NA—data not available).

Authors of the Study	Analyzed Parameters	Number of Patients	Sensitivity	Specificity	Other
Nagai et al. [47].	Age, LDH, MRI, endometrial cytology finding	63 patients with suspected uterine sarcoma (after pathological examination: 15 with uterine sarcoma and 48 with benign tumors)	80%	85.4%	
Köhler et al. [12].	Bleeding symptoms: intermenstrual bleeding, hypermenorrhea, dysmenorrhea, postmenstrual bleeding, suspicious sonography and the tumor diameter	826 patients with LM, 239 patients with LMS	87.5%	94.23%	
Nishigaya et al. [48].	Preoperative serum concentrations of LDH, D-dimer, CRP	69 cases of LM, 36 cases of LMS and 28 cases of presumed malignancy	35.3%	100% (when all parameters were positive)	
Thomassin-Naggara et al. [87].	Clinical parameters: the age of the patient, menopausal status and MRI parameters including high b1000 signal intensity, intermediate T2-weighted signal intensity, mean, intra-tumoral hemorrhage, endometrial thickening, T2-weighted signal heterogeneity, heterogeneous enhancement and non-myometrial origin	51 patients, including 15 patients with LM and 3 patients with LMS	NA	NA	
Zhang Fenfen al [39].	Preoperative serum concentrations of CA125, LDH, HE4	37 participants with LMS and 102 participants with DUF	68.4%	95.1%	
Goto et al. [45].	LDH and MRI	227 patients including 10 patients with LMS and 130 patients with ULM and 17 patients with DULM	100%	99.2%	
Zhang Guorui et al. [49].	Age ≥ 40 years old, tumor size ≥ 7 cm, neutrophil-to-lymphocyte ratio (NLR) ≥ 2.8, number of platelet ≥ 298 × 10^9^/L and lactate dehydrogenase (LDH) ≥ 193 U/L	45 ULMS patients and 180 uterine fibroid patients	80% (in the cut-off value—4 points)	77.8% (in the cut-off value—4 points)	
Lawlor et al. [113].	Postmenopausal status, symptoms of pressure, postmenopausal bleeding, neutrophil count ≥ 7.5 × 10^9^, hemoglobin level < 118 g/L, endometrial biopsy results of cellular atypia or neoplasia, and a mass size of ≥10 cm on radiological imaging	190 patients including 159 ULM and 31 ULMS	NA	NA	
Wang et al. [118].	Three clinical parameters: the age, menopausal status, and symptoms and seven radiomic features	134 patients including 81 with LM and 53 malignant uterine mesenchymal tumors	NA	NA	The AUC value was the highest when the radiomics model was combined with clinical model to 0.91 ± 0.05 (*p* < 0.05)

## 6. Discussion

The analysis of the literature in the field led to the rise of multiple questions: Which of the methods proposed in our review is the most effective? May AI and machine learning models facilitate preoperative differentiation of ULM and ULMS, which, as we know, is one of the current challenges for clinicians and researchers? Despite the passage of time, the diagnosis of ULMS in the majority of cases is still made postoperatively. It is substantial not only for the patients and the prognosis, but also for clinicians, who face the choice of the most adequate therapy. According to FDA recommendations, morcellation should be used only with a containment system in the cases of lesions, which have been suspected of being benign [9]. This guideline is a result of George et al.’s study, in which the authors have observed that the use of morcellation in the cases where presumed ULM was associated with over a three-fold increased risk of recurrence and significantly shorter median recurrence-free survival (10.8 months vs 39.6 months) [9]. The knowledge about the consequences of the use of morcellation are well known, but sometimes it is easy to forget about difficulties for pathologists, who should stage and grade the disease in order to treat it appropriately [11]. Therefore, the preoperative diagnosis of ULMS, despite of the rarity of this type of tumor, has such a big significance.

The usage of only one parameter or one diagnostic method is not sufficient for a proper diagnosis. In order to distinguish ULM and ULMS we should not only rely on an elevated concentration of one of the laboratory measurements, even if there are some studies which confirm that an increase in certain parameters may be associated with sarcomas. Taking into consideration Skorstad et al.’s study, in which the correlation between CA 125 levels and the stage of the LMS has been described, clinicians could suspect malignancy based on the analysis of only one parameter [20]. On the other hand, the study of Yilmaz et al. has not shown any correlation between CA 125 levels and uterine sarcoma [37]. Similar conclusions have been drawn regarding the LDH and other tumor markers measurements, whose elevated levels can be observed more frequently in ULMS compared to ULM. However, a juxtaposition of two or more laboratory features appeared to be promising.

Some of the above-mentioned studies have shown that more precise preoperative diagnosis could be realistic, even without the usage of imaging methods. Nishigaya et al. and Zhang Fenfen et al. used only easily accessible laboratory tests in their scales, including CA125, LDH, HE4, D-dimer and CRP. In the study of Nishigaya et al., in case all included parameters were positive, despite the low score of the sensitivity—35%, the specificity was 100% [48]. The Zhang Fenfen et al. study, due to serum concentrations of CA125, LDH, HE4 measurement, achieved the sensitivity of 68.4% and the specificity of 95.1% [39]. From our standpoint, their results are encouraging, especially taking into consideration the fact that these parameters are affordable and easy to examine. We cannot forget that a sensitivity of 68.4% is connected to one-third of cases being misdiagnosed. On the other hand, if we can diagnose easily two-thirds of patients with uterine masses, it still seems worth using. In addition, if this scale was combined with one of the imaging methods, it would probably increase its efficacy.

Among all imaging methods, MRI, especially DWI, due to the possibility of showing better morphologic information of the soft tissue intensity, seems to be the most accurate method with regard to the aim of ULM and ULMS differentiation. Its effectiveness has been described by many authors: Tasaki et al. [84], Li et al. [85], Namimoto et al. [86]. In our opinion, radiomics show great potential for the future. It has been shown recently that radiomics augment radiological diagnosis in renal cancer, especially in distinguishing clear cell and non-clear cell renal cell carcinomas [118], and can help in gliomas and brain metastases differentiation [119]. From our perspective, on the grounds of the rarity of uterine sarcomas occurrence, the access to the computer-aided diagnosis systems seems to be fundamental in the near-future. We cannot rule out the argument that the access to MRI or other imaging methods is still limited. In addition, the higher costs of the more advanced methods in comparison to ultrasound cause the lack of MRI images being taken of some patients.

Some of the researchers enriched clinical and/or laboratory parameters with imaging methods. Köhler et al. created the score with the sensitivity of 87.5% and the specificity of 94.23% basing on symptoms and two image parameters [12]. Zhang Guorui et al. observed that the most important predictors of LMS including clinical features: the age ≥ 40 years old; laboratory tests: LDH ≥ 193 U/L, NLR ≥ 2.8 and number of platelet ≥ 298 × 109/L and the imaging measurement: tumor size ≥ 7 cm can be determined effortlessly and used in daily medical practice [49]. The scales of the aforementioned studies by Köhler et al. and Zhang Guorui et al. seem to be easy to implement, as the appraisement of the tumor’s diameter with the use of imaging methods is simple to perform.

According to the European Society of Gynecological Oncology Statement on Fibroid and Uterine Morcellation, preoperative endometrial biopsy should be mandatory before the surgery [10]. Halasaka et al. suggest that it is especially important in peri- and postmenopausal age with predicted morcellation [28]. Their effectiveness in identification of ULMS, which is localized in the deep, less accessible muscular myometrial layer of the uterus, varies from 51.5% to 64%. (38,107,108). On the other hand, the biopsy of the endometrium seems to be obligatory to exclude endometrial cancer before surgery, as we know the uterine myomas or sarcomas can mask symptoms of other gynecological diseases. Taking into consideration all pros and cons of histological and molecular aspects, it may be good medical practice to conduct an endometrial biopsy or curettage before an operation, however, it is not a reliable way to differentiate between LM and LMS.

In sum, preoperative differentiation methods, especially those which include more than one parameter, seem to be promising with regard to ULM and ULMS differentiation. The majority of measurements used in the scales are cheap, easily accessible and used in daily medical practice. Additionally, many hopes are connected to new technologies, especially AI, which seems to be becoming more precise and has higher efficacy, even compared to experienced clinicians. We cannot disregard the limitations of the aforementioned studies, including the fact that all of them were retrospective and, in consequence, it is difficult to estimate their efficiency in preoperative lesion identification. In addition, the number of patients with ULMS, due to the rarity of ULMS occurrence, was small and, in several studies, below ten. One of the solutions could be to create a new database of patients with sarcomas, which may help in testing available scales, preparing new algorithms, and to study how AI may change the rules of benign and malignant uterine lesions preoperative differentiation.

Unfortunately, at this stage we cannot indicate a specific scale or method to distinguish ULM and ULMS preoperatively and histopathologic evaluation of surgical tissue is still the gold standard for ULMS diagnosis. That does not change the fact that, for the purpose of assessment the effectiveness of the scales proposed by researchers, they should be investigated on a greater number of patients and in more gynecological centers.

## 7. Conclusions

The review of current literature led to the conclusion that we still have no reliable method to distinguish leiomyomas and leiomyosarcomas preoperatively. The studies in this field have, in most cases, retrospective design and the number of cases seems to be insufficient to achieve conclusive results. The rarity of malignant myometrial lesions is an obvious reason that explains such a situation.

However, on the horizon, we can identify new game-changing options that have the potential to revolutionize our practice in cases of myometrial lesions.

On one hand, there are many ambitious researchers introducing novel multi-parametric diagnostic scales, having promising performances in differentiating such lesions.

On the other hand, the recent advancement of machine learning and artificial intelligence constitute promising novel tools that give hope that we can solve this problem completely. Nowadays, the biggest challenge for researchers and clinicians is organizing multicenter databases that include numerous cases of both LM and LMS, and using them for the development and validation of these novel tools.

## Data Availability

Not applicable.

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
