# Peer review of "Preoperative Differentiation of Uterine Leiomyomas and Leiomyosarcomas: Current Possibilities and Future Directions"

_cancers, 2022, doi:10.3390/cancers14081966_

Round 1

Reviewer 1 Report

In their manuscript „Preoperative differentiation of uterine leiomyomas and leiomyosarcomas. Current possibilities and future directions.“ Zak and colleagues review the existing literature on the before mentioned topic, showing the current research on specific liquid-biopsy based tumor marker, imaging methods like US, CT or MRI and molecular diagnostics. The authors conclude that none of the current methods offers a reliable preoperative distinguishment of leiomyomas and leiomyosarcomas, but that radiomics and the use of artificial intelligence hold great promises in the future.

The manuscript itself is written mainly understandable, and covers a topic of high clinical interest. With 119 references, it is a very comprehensive review of the literature. However, at some points, it would be beneficial to sharpen the manuscript. I have some minor concerns on the article, which are here in detail.

  • Although the manuscript is written understandable, there are some strange sounding phrases, typos and some over-complicated sentences in it. For instance: l. 37 and other: passage of time; l.138: widely used; l. 157: to be not useful; l.267: that type; l. 280 – 282: this sentence sounds strange; l. 285: USS?; l. 585: to avoid telomerase -> sentences seems truncated – avoid what: telomerase activity decline?; and some more. I would recommend proof-reading of the whole manuscript.
  • Under 2.2 specific markers, the subheadings are labelled wrong (3.2.1 – 3.2.4). Furthermore, as in 3.2.3 only markers are described, which did not show a correlation to ULM/ULMS distinguishment, I would recommend to evaluate, whether the information of the individual subchapters can be joined and Table 3 could be omitted.
  • In Table 4, the column “Main findings” is missing. In general, I like the similar style of the individual tables. I would also recommend to sharpen the points in the columns “Main findings”.
  • It could be evaluated, whether the subchapters 4.1. and 4.2. can be fused.

Finally, I would like to thank the authors for compiling the existing literature on this topic for the scientific community. Best regards.

Reviewer 2 Report

Dear Authors,

Thank you for the opportunity to review this interesting article. This is a very interesting review article with emphasis on uterine leiomyomas.

Although the review article is composed of relevant information and relevant literature, the reviewer has some observations that the authors may want to consider:

Line 14: Rephrase the sentence "can impact on the choose of treatment method";

Line 25: Rephrase the sentence "of ULMs and ULM preoperative differentiation ULMs and ULM";

Line 28: Rephrase "we can base on AI";

Line 42: Correct "Lasty";

Lines (all lines) below: The manuscript needs a grammar check, preferably by a native speaker. However, it seems that the majority of paragraphs are written clearly and correctly, but a language check is mandatory;

CA125, LDH and other markers studies: Discuss about the power of these studies due to the their sample sizes, which are small.

Materials and methods: Please describe how the search terms were constructed, eg. if the boolean operators were used.
